# Task Descriptors Help Transformers Learn Linear Models In-Context

**Ruomin Huang** [1]   **Rong Ge** [1]

## Abstract

Large language models (LLM) exhibit strong in-context learning (ICL) ability, which allows the model to make predictions on new examples based on the given prompt. Recently, a line of research (Von Oswald et al., 2023; Akyürek et al., 2023; Ahn et al., 2023; Mahankali et al., 2023; Zhang et al., 2024) considered ICL for a simple linear regression setting and showed that the forward pass of Transformers is simulating some variants of gradient descent (GD) algorithms on the in-context examples. In practice, the input prompt usually contains two types of information: in-context examples and the task description. Therefore, in this research, we will try to theoretically investigate how the task description helps ICL. Specifically, our input prompt contains not only in-context examples but also a "task descriptor".We empirically show that the trained transformer can achieve significantly lower loss for ICL when the task descriptor is provided. We further give a global convergence theorem, where the converged parameters match our experimental result.

## 1. Introduction

In recent years, Transformer-based large language models (LLM) have exhibited surprising abilities. One of the most remarkable abilities is to perform well universally, even for the tasks that they are not explicitly trained on. This is partially attributed to in-context learning (ICL) mechanism, where in-context examples are provided to significantly improve the prediction of LLM on a new query input (Brown et al., 2020).

To obtain a better understanding of ICL mechanism, the problem of learning a function class $\mathcal{H}$ in-context is proposed (Garg et al., 2022). Specifically, given an input se-

quence $S = (x_1, h(x_1), \ldots, x_n, h(x_n), x_{\text{query}})$, the model $f$ can output a prediction $f(S)$. Here the data $x_i, x_{\text{query}}$ are i.i.d. samples from an underlying distribution $D_{\mathcal{X}}$ and ground truth function $h$ is drawn from a distribution $D_{\mathcal{H}}$ over functions in $\mathcal{H}$. We say the model $f$ in-context learn the function class $\mathcal{H}$ up to $\epsilon$, if we have the expected loss $\mathbb{E}_{x_i, x_{\text{query}}, h}[\ell(f(S), h(x_{\text{query}}))] \leq \epsilon$ for large enough $n$, where $\ell(\cdot, \cdot)$ is some loss function, e.g., the mean square error. It has been observed that Transformers can in-context learn linear models. Several studies have followed this line of research to explore the mechanism of ICL by solving least-square linear regressions (Von Oswald et al., 2023; Akyürek et al., 2023; Ahn et al., 2023; Mahankali et al., 2023; Zhang et al., 2024). In their works, the input prompts take the form $(x_1, w^\top x_1, x_2, w^\top x_2, \ldots, x_n, w^\top x_n, x_{\text{query}})$ where $x_i, x_{\text{query}}$ are i.i.d. samples from some Gaussian distribution $\mathcal{N}(\mu, \Lambda)$ and $w$ is independently sampled from $\mathcal{N}(0, I_d)$. It was then proposed that there are some specific parameters under which one forward pass of Transformers is equivalent to one step of some variant of gradient descent of a linear model (Von Oswald et al., 2023; Ahn et al., 2023; Mahankali et al., 2023). Zhang et al. (2024) investigated how Transformers can be trained to exhibit ICL ability by proving that single-layer linear Transformers with appropriate initialization, will converge to the global minimum under the gradient flow dynamics. These works revealed that trained Transformers can express universal algorithms, such as variants of gradient descent.

While the existing analysis relies purely on in-context examples, is that the only information in context? In practice, the context usually contains task descriptions. For instance, one may explicitly instruct LLM to translate before giving English-French pairs (see Figure 1).

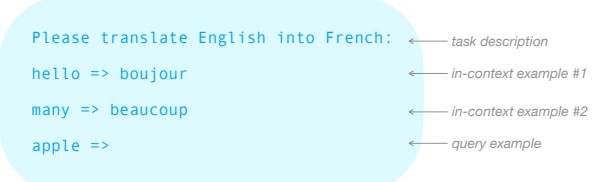

*Figure 1.* An input with both task descriptions and in-context examples.

It has been widely observed that models can make use of

---

[1]Duke University. Correspondence to: Rong Ge <rongge@cs.duke.edu>.

*Proceedings of the 1st Workshop on In-Context Learning at the 41st International Conference on Machine Learning*, Vienna, Austria. 2024. Copyright 2024 by the author(s).

natural language task descriptions to better perform ICL (Brown et al., 2020). For example, by adding a token indicating which domain the data comes from, LLM can learn knowledge from the context more efficiently (Allen-Zhu & Li, 2024). In this paper, we will investigate how Transformers can leverage task descriptions in context. Specifically, our input prompt contains not only in-context examples but also a "task descriptor" for each task $\tau$.

## 2. Setup: Mean-Varying Linear Regressions

Following the previous line of work (Von Oswald et al., 2023; Akyürek et al., 2023; Ahn et al., 2023; Mahankali et al., 2023; Zhang et al., 2024), we introduce the *mean-varying linear regressions* problem. For each linear regression task $\tau$, we independently sample in-context examples $x_{\tau,i}$ and the query example $x_{\tau,\text{query}}$ from some Gaussian distribution $\mathcal{N}(\mu_\tau, \Lambda)$. Here the mean $\mu_\tau$ is $\tau$-dependent and $\mu_\tau$ is independently sampled from $\mathcal{N}(0, I_d)$ for each task $\tau$. The covariance matrix $\Lambda$ is a diagonal matrix independent of $\tau$. A valid descriptor for above defined task is the mean $\mu_\tau$. There are many other variants of task descriptors, but as we will see later, this setting allows efficient ICL with linear self-attention. Therefore, input sequence with task descriptors is $S_\tau = (\mu_\tau, x_{\tau,1}, w_\tau^\top x_{\tau,1}, \ldots, x_{\tau,n}, w_\tau^\top x_{\tau,n}, x_{\tau,\text{query}})$. Specifically, input sequence $S_\tau$ can be generated from the following process:

**Input sequence with descriptors for task $\tau$.**

1. Draw the input mean $\mu_\tau$ from $\mathcal{N}(0, I_d)$;

2. Draw the weight vector $w_\tau$ from $\mathcal{N}(0, I_d)$;

3. For $i = 1, \ldots, n$, draw $x_{\tau,i}$ from $\mathcal{N}(\mu_\tau, \Lambda)$;

4. Return the sequence

$$S_\tau = (\mu_\tau, x_{\tau,1}, w_\tau^\top x_{\tau,1}, \ldots, x_{\tau,n}, w_\tau^\top x_{\tau,n}, x_{\tau,\text{query}}).$$

**Embedding matrix $E_\tau$.** Since it is flexible to construct the input embedding matrix $E_\tau$ from the input token sequence $S_\tau$, in this paper, we consider the following embedding matrix $E_\tau$ which duplicates the task descriptor before each stack of $(x, y)^\top$. That is,

$$E_\tau = \begin{pmatrix} \mu_\tau & \mu_\tau & \ldots & \mu_\tau & \mu_\tau \\ x_{\tau,1} & x_{\tau,2} & \ldots & x_{\tau,n} & x_{\tau,\text{query}} \\ y_{\tau,1} & y_{\tau,2} & \ldots & y_{\tau,n} & 0 \end{pmatrix}. \quad (1)$$

Here we set the last query stack to be $(\mu_\tau, x_{\tau,\text{query}}, 0)^\top$ and the zero entry remains to be filled with the prediction of the model. [1]

---

[1]Note that the format of the embedding matrix is flexible, hence it is not necessary to duplicate task descriptors and pair $x, y$ as a stack.

**Model architecture.** The softmax self-attention Transformer is

$$f(E; W) = E + W^P W^V E \cdot \text{softmax}\left(\frac{E^\top W^K W^Q E}{\rho}\right)$$

where $\rho$ is a normalizing factor and $E$ is the input embedding matrix. In this paper, we will consider a simplified version of one-layer linear self-attention (LSA) Transformer, which is adopted from Zhang et al. (2024). Specifically, the projection matrix and the value matrix are merged into a projection-value matrix $W^{PV} \in \mathbb{R}^{d \times d}$, and the key matrix and query matrix are merged into a key-query matrix $W^{KQ} \in \mathbb{R}^{d \times d}$:

$$f_{\text{LSA}}(E; W) = E + W^{PV} E \cdot \frac{E^\top W^{KQ} E}{n}. \quad (2)$$

Here $W = (W^{KQ}, W^{PV})$ and the normalizing factor is set to be the number of in-context examples $n$. For the input with task descriptors $E = E_\tau$, the prediction is given by the right-bottom entry $\hat{y}_{\tau,\text{query}} = f_{\text{LSA}}(E_\tau; W)_{2d+1,n+1}$.

**Initialization.** We make the following assumption on the initialization. The assumption is motivated by the initialization in Zhang et al. (2024).

**Assumption 2.1** (Initialization). We assume the initialization of the Transformer satisfies

$$W^{KQ}(0) = \begin{pmatrix} \Sigma_{11} & \Sigma_{12} & 0_d \\ \Sigma_{21} & \Sigma_{22} & 0_d \\ 0_d^\top & 0_d^\top & 0 \end{pmatrix}, W^{PV}(0) = \begin{pmatrix} 0_{d \times d} & 0_{d \times d} & 0_d \\ 0_{d \times d} & 0_{d \times d} & 0_d \\ 0_d^\top & 0_d^\top & \sigma \end{pmatrix}$$

where $\Sigma_{11}, \Sigma_{22}, \Sigma_{12}, \Sigma_{21}$ are PSD matrices such that the sum of the Frobenius norms

$$\sigma := \|\Sigma_{11}\|_F^2 + \|\Sigma_{12}\|_F^2 + \|\Sigma_{21}\|_F^2 + \|\Sigma_{22}\|_F^2 > 0.$$

A simple way to satisfy the requirement is to take $\Sigma_{11} = \Sigma_{12} = \Sigma_{21} = \Sigma_{22} = I_d$ and $\sigma = 4d$.

**Training procedure.** Let $\ell(W, \tau)$ be the expected least-square error for task $\tau$. That is,

$$\ell(W, \tau) := \frac{1}{2} \mathbb{E}_{x_{\tau,i}, x_{\tau,\text{query}}, w_\tau}[(\hat{y}_{\tau,\text{query}} - w_\tau^\top x_{\tau,\text{query}})^2]. \quad (3)$$

We consider the population loss

$$L(W) := \mathbb{E}_{\mu_\tau \sim \mathcal{N}(0, I_d)}[\ell(W, \tau)] \quad (4)$$

and our training algorithm is the gradient flow:

$$\frac{dW}{dt} = -\nabla L(W). \quad (5)$$

In next section, we will both theoretically and empirically show that task descriptors help Transformers learn mean-varying linear regressions in-context.

# 3. Task Descriptors Help Transformers Learn In-Context

We investigate both the optimal parameters $W_*$ and the training dynamics of gradient flow (5). Our main result can be summarized as the following theorem.

**Theorem 3.1** (Main result). *Under Assumption 2.1, if the number of samples $n \to \infty$ and $\sigma$ satisfies $0 < \sigma < \alpha$ for some contant $\alpha$* [2]*, then the gradient flow (5) will converge*[3] *to the global minimizer $W_* = (W_*^{KQ}, W_*^{PV})$ and the corresponding loss $\lim_{n \to \infty} L(W_*) = 0$. Here we have*

$$
W_*^{KQ} = \begin{pmatrix} 0_{d \times d} & -\frac{1}{w^*}\Lambda^{-1} & 0_d \\ 0_{d \times d} & \frac{1}{w^*}\Lambda^{-1} & 0_d \\ 0_d^\top & 0_d^\top & 0 \end{pmatrix} \tag{6}
$$

*and*

$$
W_*^{PV} = \begin{pmatrix} 0_{d \times d} & 0_{d \times d} & 0_d \\ 0_{d \times d} & 0_{d \times d} & 0_d \\ 0_d^\top & 0_d^\top & w^* \end{pmatrix} \tag{7}
$$

*where $w^* = \left(2\|\Lambda^{-1}\|_F^2\right)^{\frac{1}{4}}$.*

*Remark* 3.2. Theorem 3.1 is proved by showing a error bound (Luo & Tseng, 1993) of population loss (4), which is presented in Lemma A.3. Noting that scaling $W^{KQ}$ by a factor $\rho$ and scaling $W^{PV}$ by $1/\rho$ will not affect the output, which implies there are infinite global minimizers. Hence we show that $W^{PV}$ and $W^{KQ}$ are balanced in Lemma A.1, which implies that gradient flow converges to the balanced minimizer among infinite minimizers.

**Standardization Operator**  Comparing Theorem 3.1 with the result of Zhang et al. (2024), we found that when receiving the input with the mean $\mu_\tau$ as the task description component, well-trained Transformers will perform an additional "standardization" operator

$$
C = \begin{pmatrix} 0_{d \times d} & 0_{d \times d} & 0_d \\ -I_d & I_d & 0_d \\ 0_d^\top & 0_d^\top & 1 \end{pmatrix} \in \mathbb{R}^{(2d+1) \times (2d+1)} \tag{8}
$$

on the key matrix in the Theorem 4.1 of Zhang et al. (2024). Specifically, letting $\tilde{W}_*^{KQ}$ and $\tilde{W}_*^{PV} \in \mathbb{R}^{(d+1) \times (d+1)}$ be the converged key-query matrix and projection-value matrix in the Theorem 4.1 of Zhang et al. (2024). We have

$$
W_*^{KQ} = C^\top \begin{pmatrix} 0_{d \times d} & 0_{d \times (d+1)} \\ 0_{(d+1) \times d} & \tilde{W}_*^{KQ} \end{pmatrix} \tag{9}
$$

and

$$
W_*^{PV} = \begin{pmatrix} 0_{d \times d} & 0_{d \times (d+1)} \\ 0_{(d+1) \times d} & \tilde{W}_*^{PV} \end{pmatrix}. \tag{10}
$$

---

[2]Please see Lemma A.2 in the appendix for the value of $\alpha$.
[3]Here the gradient flow becomes $\frac{dW}{dt} = -\nabla \lim_{n \to \infty} L(W)$.

This observation implies that provided with $\mu_\tau$ in the task descriptions, well-trained Transformers will first use the task descriptions to standardize the input data and then perform one step of preconditioned GD (Ahn et al., 2023; Zhang et al., 2024), which is a natural way to convert the current task into the "standard" task using task descriptions.

Now we give some calculations showing why $W_*$ works and why task descriptors are needed. For an input embedding matrix $E_\tau$, denote $\bar{E}_\tau := CE_\tau$ the standardized embedding matrix and $\bar{x} := x - \mu_\tau$ the standardized data. Then we have

$$
\bar{E}_\tau = \begin{pmatrix} 0 & 0 & \cdots & 0 & 0 \\ \bar{x}_1 & \bar{x}_2 & \cdots & \bar{x}_n & \bar{x}_{\text{query}} \\ y_1 & y_2 & \cdots & y_n & 0 \end{pmatrix}. \tag{11}
$$

Then we perform one step of preconditioned GD on the standardized data to get the prediction $\hat{y}_{\text{query}}$. Here by Theorem 4.1 of Zhang et al. (2024), the preconditioner is $\Lambda^{-1}$ if $n$ goes to infinity. Therefore we have

$$
\begin{aligned}
\hat{y}_{\text{query}} &= x_{\text{query}}^\top \Lambda^{-1} \frac{1}{n} \sum_{i=1}^n \bar{x}_i y_i \\
&= x_{\text{query}}^\top \Lambda^{-1} \left( \frac{1}{n} \sum_{i=1}^n \bar{x}_i x_i^\top \right) w \\
&\to x_{\text{query}}^\top \Lambda^{-1} \Lambda w \quad \text{as } n \to \infty.
\end{aligned} \tag{12}
$$

If the input does not contain task descriptors and Transformers directly perform one step of preconditioned GD on the original data using some preconditioner $A$, the prediction is

$$
\begin{aligned}
\hat{y}_{\text{query}} &= x_{\text{query}}^\top A\left(\frac{1}{n} \sum_{i=1}^n x_i y_i\right) \\
&= x_{\text{query}}^\top A\left(\frac{1}{n} \sum_{i=1}^n x_i x_i^\top\right) w \\
&\to x_{\text{query}}^\top A\left(\Lambda + \mu_\tau \mu_\tau^\top\right) w \quad \text{as } n \to \infty.
\end{aligned} \tag{13}
$$

We can see the prediction will depend on mean $\mu$, which is problematic since in our setting $\mu_\tau$ is not fixed across input sequences. Note that if $\mu_\tau$ is fixed then a preconditioner $A = (\Lambda + \mu_\tau \mu_\tau^\top)^{-1}$ works.

# 4. Experiments

We train one-layer LSA Transformers which do not merge key-query and projection-value:

$$
f(E; W) = E + W^P W^V E \cdot \left(\frac{E^\top W^K W^Q E}{n}\right).
$$

Our weight matrices $W^P, W^V, W^K$ and $W^V$ are all $(2d+1) \times (2d+1)$ matrices. We use embedding matrices with task descriptor $E_\tau$ in (1) and embedding matrices without

task descriptors respectively. To construct embedding matrices without task descriptors, we simply replace $\mu_\tau$ with zero vector $0_d$ in (1) hence the input dimension of two embeddings are the same. In our experiments, we use Adam optimizer (Kingma & Ba, 2015) to train the one-layer LSA Transformers. We set $n = 500, d = 5$ and $\Lambda = I_d$. We generate 2048 i.i.d. input sequences for each episode of training. We train each Transformer for 1000 epochs.

We plot the ICL loss curves during training in Figure 2, which shows there is a separation between the Transformers trained with and without task descriptors.

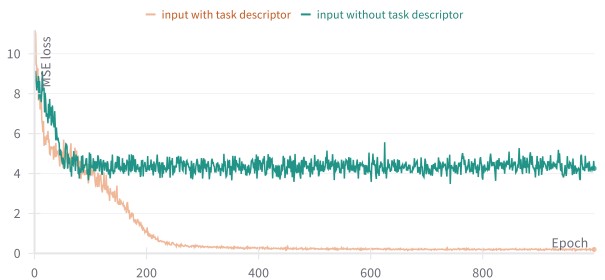

*Figure 2.* The training mean squared error for one-layer LSA Transformers .

We also plot heat maps of the weight matrices $W^{KQ}$ and $W^{PV}$ of the well-trained Transformers with task descriptors in the input (see Figure 3). From Figure 3 we can see there is a clear pattern in the $W^{KQ}$ and also a non-trivial value in the right-bottom entry of $W^{PV}$ which matches our global convergence result Theorem 3.1. It is worth noting that the prediction $\hat{y}_{\text{query}}$ only depends on the last row of $W^{PV}$ and the first $2d$ columns of $W^{KQ}$, hence the pattern looks random in $W^{PV}$ except for the last row.

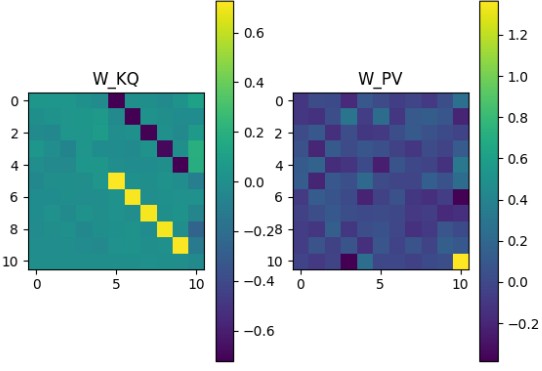

*Figure 3.* The heat map of $W^{KQ}$ and $W^{PV}$ for a well-trained Transformer with task descriptors $\mu_\tau$ in the training sequences.

## 5. Conclusions and Limitations

In this work, we investigate how Transformers leverage task descriptions in-context by adding task descriptors concatenated to the input embedding matrices. Specifically, we consider the mean-varying linear regression problem where the task descriptors can be set to be the mean $\mu_\tau$ for each task $\tau$. We give a global convergence result for Transformers trained with task descriptors under infinite samples. Our theoretical result shows that in the forward pass, Transformers standardize the input data using task descriptors before performing the key mapping. We empirically show that Transformers can achieve much lower loss for ICL when task descriptors are provided. We also find a clear pattern in the parameters of well-trained Transformers, which verifies our theoretical result. However, our embedding matrix duplicates the task descriptors, which might not align with the real-world scenario. Our theoretical result relies on the infinite-sample assumption. We leave resolving these limitations as future work.

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

# A. Omitted proofs

## A.1. Proof Sketch

Here we give the sketch of our proof to Theorem 3.1, which follows the proof framework in Zhang et al. (2024). Before we start, let's write $W^{PV}$ and $W^{KQ}$ into blocks:

$$W^{PV} = \begin{pmatrix} W_{11}^{PV} & W_{12}^{PV} & w_{13}^{PV} \\ W_{21}^{PV} & W_{22}^{PV} & w_{23}^{PV} \\ (w_{31}^{PV})^\top & (w_{32}^{PV})^\top & w_{33}^{PV} \end{pmatrix} \tag{14}$$

and

$$W^{KQ} = \begin{pmatrix} W_{11}^{KQ} & W_{12}^{KQ} & w_{13}^{KQ} \\ W_{21}^{KQ} & W_{22}^{KQ} & w_{23}^{KQ} \\ (w_{31}^{KQ})^\top & (w_{32}^{KQ})^\top & w_{33}^{KQ} \end{pmatrix} \tag{15}$$

where all the $W_{11}, W_{12}, W_{21}, W_{22} \in \mathbb{R}^{d \times d}$, $w_{13}, w_{23}, w_{31}, w_{32} \in \mathbb{R}^d$ and $w_{33} \in \mathbb{R}$. By expanding the prediction $\hat{y}_{\tau,\text{query}} = f_{\text{LSA}}(E_\tau; W)_{2d+1,n+1}$, we know the prediction only depends on the weight blocks $W_{11}^{KQ}, W_{12}^{KQ}, w_{31}^{PV}, W_{21}^{KQ}, W_{22}^{KQ}, w_{32}^{PV}, w_{31}^{KQ}, w_{32}^{KQ}$ and $w_{33}^{PV}$. Therefore we will only consider the training dynamics of these relevant blocks. To simplify notation we gather all the relevant parameters in the following block matrix $U$.

$$\begin{pmatrix} U_{11} & U_{12} & u_{13} \\ U_{21} & U_{22} & u_{23} \\ u_{31}^\top & u_{32}^\top & u_{-1} \end{pmatrix} := \begin{pmatrix} W_{11}^{KQ} & W_{12}^{KQ} & w_{31}^{PV} \\ W_{21}^{KQ} & W_{22}^{KQ} & w_{32}^{PV} \\ (w_{31}^{KQ})^\top & (w_{32}^{KQ})^\top & w_{33}^{PV} \end{pmatrix}. \tag{16}$$

We start with the dynamics of $u_{13}, u_{23}, u_{31}$ and $u_{32}$, which shows that these parameters stick to $0$ during the training phase so the dynamics of $U$ could be simplified. Then we show that there is a balance between $u_{-1}$ and $U_{11}, U_{12}, U_{21}, U_{22}$. Specifically, we have the following lemma.

**Lemma A.1.** *If our initialization satisfies Assumption 2.1, then we have both*

$$u_{13}(t) = u_{23}(t) = u_{31}(t) = u_{32}(t) = 0 \tag{17}$$

*and*

$$u_{-1}(t)^2 = \|U_{11}(t)\|_F^2 + \|U_{12}(t)\|_F^2 + \|U_{21}(t)\|_F^2 + \|U_{22}(t)\|_F^2 \tag{18}$$

*for all $t \geq 0$.*

Given the balanced condition, we can prove $u_{-1}$ could be lower bounded by some positive constant during the training phase in Lemma A.2, which suggests the trajectory of $u_{-1}$ is away from the saddle point $u_{-1} = 0$.

**Lemma A.2.** *If our initialization satisfies Assumption 2.1, $n \to \infty$ and $\sigma$ satisfies $0 < \sigma < \alpha$ where $\alpha$ is equal to*

$$\left( \frac{d+2}{2\|\Lambda\|_F \left(\|\Lambda\|_F^2 + 2\operatorname{tr}(\Lambda) + 3d^2\right) + 28d\operatorname{tr}(\Lambda) + 60d^3} \right)^{\frac{1}{2}}, \tag{19}$$

*then we have $u_{-1}(t) \geq \beta > 0$ for all $t \geq 0$. Here*

$$\beta = \frac{(d+2)\sigma}{\left(4 + 2\sqrt{2}\right)\left(\|\Lambda\|_F^2 + 2\operatorname{tr}(\Lambda) + d^2 + 2d\right)}. \tag{20}$$

With the lower bound $\beta$ of $u_{-1}$, we are finally able to give an error bound (Luo & Tseng, 1993) of our loss $L(U)$ in Lemma A.3, which is the main lemma of this work.

**Lemma A.3.** *Let $\bar{A} := \frac{1}{2}(A + A^\top)$ for any real square matrix $A$. If our initialization satisfies Assumption 2.1 and $n \to \infty$, then we have*

$$\|\nabla L(U)\|_F^2$$
$$\geq c \left( \left\|U_{11} + U_{12} + \bar{U}_{22} + \bar{U}_{21}\right\|_F^2 + \left\|U_{22} + U_{21} - \frac{\Lambda^{-1}}{u_{-1}}\right\|_F^2 \right.$$
$$\left. + \|U_{12} + \frac{\Lambda^{-1}}{u_{-1}}\|_F^2 + \|U_{22} - \frac{\Lambda^{-1}}{u_{-1}}\|_F^2 \right) \tag{21}$$

*where*

$$c = \beta^2 \min\left(\frac{\lambda_{\min}(\Lambda)^2}{30d}, \frac{1}{30d}, \frac{\lambda_{\min}(\Lambda)^4}{10}, \frac{1}{10}\right).$$

With Lemma A.3 in hand, we can finally prove Theorem 3.1.

*Proof of Theorem 3.1.* Since $L(U) \geq 0$ is bounded below, we know $L(U_t)$ the loss over gradient flow will converge. Any stationary point $U^*$ of the gradient flow must satisfy $\nabla L(U^*) = 0$. Therefore, combining with the error bound (21) we have $\|U_{22}^* + U_{21}^* - \frac{\Lambda^{-1}}{u_{-1}^*}\|_F^2 = \|U_{11}^* + U_{12}^* + \bar{U}_{22}^* + \bar{U}_{21}^*\|_F^2 = \|U_{12}^* + \frac{\Lambda^{-1}}{u_{-1}^*}\|_F^2 = \|U_{22}^* - \frac{\Lambda^{-1}}{u_{-1}^*}\|_F^2 = 0$, which implies that $U_{22}^* = \frac{\Lambda^{-1}}{u_{-1}^*}, U_{12}^* = -\frac{\Lambda^{-1}}{u_{-1}^*}, U_{21}^* = 0_{d \times d}$ and $U_{11}^* = 0$. Finally by direct computation we know the corresponding loss is $L(U^*) = 0$, which implies that $U^*$ is a global minimizer. Combining (18), we have $u_{-1}^* = \left(2\|\Lambda^{-1}\|_F^2\right)^{\frac{1}{4}}$. Translating $U$ back to $W$ according to (16), we obtain Theorem 3.1. $\qquad\square$

*Proof of Equation* (17) *in Lemma A.1.* The gradient of the loss is $\frac{\partial \ell(U, \tau)}{\partial U} = \mathbb{E}[(\hat{y}_{\tau,\text{query}} - w_\tau^\top x_{\tau,\text{query}})\frac{\partial \hat{y}_{\tau,\text{query}}}{\partial U}]$.

To give the detailed gradient formulation, we need to expand $\hat{y}_{\tau,\text{query}}$ in terms of $U$ first. Denote $\hat{\Lambda}_\tau = \frac{1}{n}\sum_{i=1}^n x_{\tau,i} x_{\tau,i}^\top$ and $\hat{\mu}_\tau = \frac{1}{n}\sum_{i=1}^n x_{\tau,i}$. Then we have

$$
\begin{aligned}
\hat{y}_{\tau,\text{query}} &= \begin{pmatrix} u_{13}^\top & u_{23}^\top & u_{-1} \end{pmatrix} \begin{pmatrix} (1+\frac{1}{n})\mu_\tau\mu_\tau^\top & \mu_\tau\hat{\mu}_\tau^\top + \frac{1}{n}\mu_\tau x_{\tau,\text{query}}^\top & \mu_\tau \cdot w_\tau^\top \hat{\mu}_\tau \\ \hat{\mu}_\tau\mu_\tau^\top + \frac{1}{n}x_{\tau,\text{query}}\mu_\tau^\top & \hat{\Lambda} + \frac{1}{n}x_{\tau,\text{query}}x_{\tau,\text{query}}^\top & \hat{\Lambda}w_\tau \\ \mu_\tau^\top \cdot w_\tau^\top \hat{\mu}_\tau & w_\tau^\top\hat{\Lambda} & w_\tau^\top\hat{\Lambda}w_\tau \end{pmatrix} \begin{pmatrix} U_{11} & U_{12} \\ U_{21} & U_{22} \\ u_{31}^\top & u_{32}^\top \end{pmatrix} \begin{pmatrix} \mu_\tau \\ x_{\tau,\text{query}} \end{pmatrix} \\
&= u_{13}^\top\left((\frac{1}{n}+1)\mu_\tau\mu_\tau^\top U_{11} + \left(\mu_\tau\hat{\mu}_\tau^\top + \frac{1}{n}\mu_\tau x_{\tau,q}^\top\right)U_{21} + w_\tau^\top\hat{\mu}_\tau\mu_\tau u_{31}^\top\right)\mu_\tau \\
&\quad + u_{13}^\top\left((\frac{1}{n}+1)\mu_\tau\mu_\tau^\top U_{12} + \left(\mu_\tau\hat{\mu}_\tau^\top + \frac{1}{n}\mu_\tau x_{\tau,q}^\top\right)U_{22} + w_\tau^\top\hat{\mu}_\tau\mu_\tau u_{32}^\top\right)x_{\tau,\text{query}} \\
&\quad + u_{23}^\top\left(\hat{\mu}_\tau\mu_\tau^\top + \frac{1}{n}x_{\tau,\text{query}}\mu_\tau^\top U_{11} + \left(\hat{\Lambda} + \frac{1}{n}x_{\tau,\text{query}}x_{\tau,\text{query}}^\top\right)\left(U_{21} + \hat{\Lambda}w_\tau u_{31}^\top\right)\right)\mu_\tau \\
&\quad + u_{23}^\top\left(\left(\hat{\mu}_\tau\mu_\tau^\top + \frac{1}{n}x_{\tau,\text{query}}\mu_\tau^\top\right)U_{12} + \left(\hat{\Lambda} + \frac{1}{n}x_{\tau,\text{query}}x_{\tau,\text{query}}^\top\right)U_{22} + \hat{\Lambda}w_\tau u_{32}^\top\right)x_{\tau,\text{query}} \\
&\quad + u_{-1} \cdot (\mu_\tau^\top w_\tau^\top\hat{\mu}_\tau U_{11}\mu_\tau + w_\tau^\top\hat{\Lambda}U_{21}\mu_\tau + w_\tau^\top\hat{\Lambda}w_\tau u_{31}^\top\mu_\tau) \\
&\quad + u_{-1} \cdot (\mu_\tau^\top w_\tau^\top\hat{\mu}_\tau U_{12}x_{\tau,\text{query}} + w_\tau^\top\hat{\Lambda}U_{22}x_{\tau,\text{query}} + w_\tau^\top\hat{\Lambda}w_\tau u_{32}^\top x_{\tau,\text{query}}).
\end{aligned}
\tag{22}
$$

If letting $u_{13} = u_{23} = u_{31} = u_{32} = 0$, then we have

$$\hat{y}_{\tau,\text{query}} = u_{-1}(\mu_\tau^\top w_\tau^\top\hat{\mu}_\tau U_{11}\mu_\tau + w_\tau^\top\hat{\Lambda}U_{21}\mu_\tau + \mu_\tau^\top w_\tau^\top\hat{\mu}_\tau U_{12}x_{\tau,\text{query}} + w_\tau^\top\hat{\Lambda}U_{22}x_{\tau,\text{query}}). \tag{23}$$

The gradient on $u_{13}$ is

$$
\begin{aligned}
\frac{\partial \ell(U, \tau)}{\partial u_{13}} &= \mathbb{E}\left[(\hat{y}_{\tau,\text{query}} - w_\tau^\top x_{\tau,\text{query}})\left((\frac{1}{n}+1)\mu_\tau\mu_\tau^\top U_{11} + \left(\mu_\tau\hat{\mu}_\tau^\top + \frac{1}{n}\mu_\tau x_{\tau,q}^\top\right)U_{21} + w_\tau^\top\hat{\mu}_\tau\mu_\tau u_{31}^\top\right)\mu_\tau\right] \\
&= \mathbb{E}\left[(\hat{y}_{\tau,\text{query}} - w_\tau^\top x_{\tau,\text{query}})\left((\frac{1}{n}+1)\mu_\tau\mu_\tau^\top U_{11} + \left(\mu_\tau\hat{\mu}_\tau^\top + \frac{1}{n}\mu_\tau x_{\tau,q}^\top\right)U_{21}\right)\mu_\tau\right].
\end{aligned}
$$

Note that

$$\hat{y}_{\tau,\text{query}} - w_\tau^\top x_{\tau,\text{query}} = u_{-1}w_\tau^\top \cdot (\hat{\mu}_\tau\mu_\tau^\top U_{11}\mu_\tau + \hat{\Lambda}U_{21}\mu_\tau + \hat{\mu}_\tau\mu_\tau^\top U_{12}x_{\tau,\text{query}} + \hat{\Lambda}U_{22}x_{\tau,\text{query}} - \frac{x_{\tau,\text{query}}}{u_{-1}})$$

and

$$\left((\frac{1}{n}+1)\mu_\tau\mu_\tau^\top U_{11} + \left(\mu_\tau\hat{\mu}_\tau^\top + \frac{1}{n}\mu_\tau x_{\tau,q}^\top\right)U_{22}\right)\mu_\tau$$

does not contain $w_\tau$. Since $\mathbb{E}[w_\tau] = 0$ and $w_\tau$ is independent with all other random variables, we have $\frac{\partial \ell(U,\tau)}{\partial u_{13}} = 0$.

Similarly, we have $\frac{\partial \ell(U,\tau)}{\partial u_{23}} = 0$ given that $u_{13} = u_{23} = u_{31} = u_{32} = 0$.

Let $\Delta := (\hat{\mu}_\tau \mu_\tau^\top U_{11} \mu_\tau + \hat{\Lambda} U_{21} \mu_\tau + \hat{\mu}_\tau \mu_\tau^\top U_{12} x_{\tau,\text{query}} + \hat{\Lambda} U_{22} x_{\tau,\text{query}} - \frac{x_{\tau,\text{query}}}{u_{-1}}) \mu_\tau$. Then the gradient on $u_{31}$ is

$$\begin{aligned}
\frac{\partial \ell(U,\tau)}{\partial u_{31}} &= \mathbb{E}\left[ u_{-1} w_\tau^\top \hat{\Lambda}_\tau w_\tau (\hat{y}_{\tau,\text{query}} - w_\tau^\top x_{\tau,\text{query}}) \mu_\tau \right] \\
&= \mathbb{E}\left[ u_{-1}^2 w_\tau^\top \hat{\Lambda}_\tau w_\tau w_\tau^\top \Delta \right] \\
&= \mathbb{E}\left[ u_{-1}^2 \mathbb{E}_{w_\tau}[w_\tau^\top \hat{\Lambda}_\tau w_\tau w_\tau^\top] \Delta \right] \\
&= 0.
\end{aligned}$$

Similarly, we have $\frac{\partial \ell(U,\tau)}{\partial u_{32}} = 0$ given that $u_{13} = u_{23} = u_{31} = u_{32} = 0$. Taking expectation over $\mu_\tau$, we have $\frac{\partial L(U)}{\partial u_{13}} = \frac{\partial L(U)}{\partial u_{23}} = \frac{\partial L(U)}{\partial u_{31}} = \frac{\partial L(U)}{\partial u_{32}} = 0$ given that $u_{13} = u_{23} = u_{31} = u_{32} = 0$, which finishes the proof. $\qquad\square$

*Proof of Equation* (18) *in Lemma A.1.* Now we can simplify the prediction $\hat{y}_{\tau,\text{query}}$ by letting $u_{13} = u_{23} = u_{31} = u_{32} = 0$ in (22), which gives the prediction $\hat{y}_{\tau,\text{query}} = u_{-1} w_\tau^\top \left( \left( \hat{\mu}_\tau \mu_\tau^\top U_{11} + \hat{\Lambda}_\tau U_{21} \right) \mu_\tau + \left( \hat{\mu}_\tau \mu_\tau^\top U_{12} + \hat{\Lambda}_\tau U_{22} \right) x_{\tau,\text{query}} \right)$. This implies that

$$\hat{y}_{\tau,\text{query}} - y_{\tau,\text{query}} = u_{-1} w_\tau^\top \left( \left( \hat{\mu}_\tau \mu_\tau^\top U_{11} + \hat{\Lambda}_\tau U_{21} \right) \mu_\tau + \left( \hat{\mu}_\tau \mu_\tau^\top U_{12} + \hat{\Lambda}_\tau U_{22} - \frac{1}{u_{-1}} I_d \right) x_{\tau,\text{query}} \right). \tag{24}$$

Now we can compute the dynamics of $U$ by the chain rule $\frac{\partial \ell(U,\tau)}{\partial U} = \mathbb{E}\left[ (\hat{y}_{\tau,\text{query}} - y_{\tau,\text{query}}) \frac{\partial (\hat{y}_{\tau,\text{query}} - y_{\tau,\text{query}})}{\partial U} \right]$.

Therefore, we have the dynamics of $U_{11}, U_{12}, U_{21}, U_{22}$ and $u_{-1}$ as follows:

- 
$$\frac{\partial \ell(U,\tau)}{\partial U_{11}} = \mathbb{E}\left[ (\hat{y}_{\tau,\text{query}} - y_{\tau,\text{query}}) u_{-1} w_\tau^\top \hat{\mu}_\tau \mu_\tau \mu_\tau^\top \right]; \tag{25}$$

- 
$$\frac{\partial \ell(U,\tau)}{\partial U_{21}} = \mathbb{E}\left[ (\hat{y}_{\tau,\text{query}} - y_{\tau,\text{query}}) u_{-1} \hat{\Lambda}_\tau w_\tau \mu_\tau^\top \right] \tag{26}$$

- 
$$\frac{\partial \ell(U,\tau)}{\partial U_{12}} = \mathbb{E}\left[ (\hat{y}_{\tau,\text{query}} - y_{\tau,\text{query}}) u_{-1} w_\tau^\top \hat{\mu}_\tau \mu_\tau x_{\tau,\text{query}}^\top \right] \tag{27}$$

- 
$$\frac{\partial \ell(U,\tau)}{\partial U_{22}} = \mathbb{E}\left[ (\hat{y}_{\tau,\text{query}} - y_{\tau,\text{query}}) u_{-1} \hat{\Lambda}_\tau w_\tau x_{\tau,\text{query}}^\top \right] \tag{28}$$

- 
$$\frac{\partial \ell(U,\tau)}{\partial u_{-1}} = \mathbb{E}\left[ (\hat{y}_{\tau,\text{query}} - y_{\tau,\text{query}}) w_\tau^\top \left( M_2 \mu_\tau + \left( M_1 + \frac{1}{u_{-1}} I_d \right) x_{\tau,\text{query}} \right) \right]. \tag{29}$$

Here $M_1 := \hat{\mu}_\tau \mu_\tau^\top U_{12} + \hat{\Lambda}_\tau U_{22} - \frac{1}{u_{-1}} I_d$ and $M_2 := \hat{\mu}_\tau \mu_\tau^\top U_{11} + \hat{\Lambda}_\tau U_{21}$. Therefore we have

$$\frac{\partial \ell(U,\tau)}{\partial u_{-1}} \cdot u_{-1} = \text{tr}\left( U_{11}^\top \frac{\partial \ell(U,\tau)}{\partial U_{11}} + U_{12}^\top \frac{\partial \ell(U,\tau)}{\partial U_{12}} + U_{21}^\top \frac{\partial \ell(U,\tau)}{\partial U_{21}} + U_{22}^\top \frac{\partial \ell(U,\tau)}{\partial U_{22}} \right). \tag{30}$$

Taking expectation over $\mu_\tau$, we have the same thing holds for $L(U)$

$$\frac{\partial L(U)}{\partial u_{-1}} \cdot u_{-1} = \text{tr}\left( U_{11}^\top \frac{\partial L(U)}{\partial U_{11}} + U_{12}^\top \frac{\partial L(U)}{\partial U_{12}} + U_{21}^\top \frac{\partial L(U)}{\partial U_{21}} + U_{22}^\top \frac{\partial L(U)}{\partial U_{22}} \right). \tag{31}$$

This implies that

$$\frac{du_{-1}^2(t)}{dt} = \frac{d}{dt} \operatorname{tr} \left( U_{11}(t)U_1^\top(t) + U_{12}(t)U_{22}^\top(t) + U_{21}(t)U_{21}^\top(t) + U_{22}(t)U_{22}^\top(t) \right). \tag{32}$$

Therefore if we set $u_{-1}(0)^2 = \|U_{11}(0)\|_F^2 + \|U_{12}(0)\|_F^2 + \|U_{21}(0)\|_F^2 + \|U_{22}(0)\|_F^2$ at initialization, we have

$$u_{-1}(t)^2 = \|U_{11}(t)\|_F^2 + \|U_{12}(t)\|_F^2 + \|U_{21}(t)\|_F^2 + \|U_{22}(t)\|_F^2 \tag{33}$$

for all $t \geq 0$. $\qquad\square$

*Proof of Lemma A.2.* We will decompose the loss $\ell(U,\tau)$ into $\ell(U,\tau) = \ell_1(U,\tau) + \ell_2(U,\tau)$ and bound the coefficients of $u_{-1}$ in $\ell_1$ and $\ell_2$ separately. Recall $M_1 = \hat{\mu}_\tau \mu_\tau^\top U_{12} + \hat{\Lambda}_\tau U_{22} - \frac{1}{u_{-1}} I_d$ and $M_2 = \hat{\mu}_\tau \mu_\tau^\top U_{11} + \hat{\Lambda}_\tau U_{21}$. Then we have

$$\ell(U,\tau) = \frac{1}{2} \mathbb{E}[(\hat{y}_{\tau,\text{query}} - y_{\tau,\text{query}})^2] \tag{34}$$

$$= \frac{u_{-1}^2}{2} \left( \mathbb{E}\left[ \mu_\tau^\top M_2^\top w_\tau w_\tau^\top M_2 \mu_\tau \right] + \mathbb{E}\left[ x_{\tau,\text{query}}^\top M_1^\top w_\tau w_\tau^\top M_1 x_{\tau,\text{query}} \right] + 2\mathbb{E}\left[ x_{\tau,\text{query}}^\top M_1^\top w_\tau w_\tau^\top M_2 \mu_\tau \right] \right) \tag{35}$$

$$= \frac{u_{-1}^2}{2} \left( \mathbb{E}\left[ \mu_\tau^\top M_2^\top M_2 \mu_\tau \right] + \mathbb{E}\left[ x_{\tau,\text{query}}^\top M_1^\top M_1 x_{\tau,\text{query}} \right] + 2\mathbb{E}\left[ x_{\tau,\text{query}}^\top M_1^\top M_2 \mu_\tau \right] \right) \tag{36}$$

$$= \frac{u_{-1}^2}{2} \left( \mathbb{E}\left[ \operatorname{tr}\left( M_2^\top M_2 \mu_\tau \mu_\tau^\top \right) \right] + \mathbb{E}\left[ \operatorname{tr}(M_1^\top M_1 x_{\tau,\text{query}} x_{\tau,\text{query}}^\top) \right] + 2\mathbb{E}\left[ \operatorname{tr}(M_1^\top M_2 \mu_\tau x_{\tau,\text{query}}^\top) \right] \right) \tag{37}$$

$$= \frac{u_{-1}^2}{2} \left( \mathbb{E}\left[ \operatorname{tr}(M_2^\top M_2 \mu_\tau \mu_\tau^\top) \right] + \mathbb{E}\left[ \operatorname{tr}\left( M_1^\top M_1 \left( \Lambda + \mu_\tau \mu_\tau^\top \right) \right) \right] + 2\mathbb{E}\left[ \operatorname{tr}\left( M_1^\top M_2 \mu_\tau \mu_\tau^\top \right) \right] \right) \tag{38}$$

$$= \underbrace{\frac{u_{-1}^2}{2} \mathbb{E}\left[ \operatorname{tr}\left( M_1^\top M_1 \Lambda \right) \right]}_{\ell_1(U,\tau)} + \underbrace{\frac{u_{-1}^2}{2} \mathbb{E}\left[ \operatorname{tr}\left( (M_2 + M_1)^\top (M_2 + M_1) \mu_\tau \mu_\tau^\top \right) \right]}_{\ell_2(U,\tau)}. \tag{39}$$

Now we compute the expectation in $\ell_1$ and $\ell_2$. Define a positive value $\gamma = \|\mu_\tau\|^2 + \frac{1}{n} \operatorname{tr}(\Lambda)$ and a positive definite matrix $\Gamma = \frac{n+1}{n} \left( \Lambda + \mu_\tau \mu_\tau^\top \right) + \frac{1}{n} \left( \operatorname{tr}(\Lambda) + \|\mu_\tau\|^2 \right) I_d$. By direct computation we have

$$\ell_1(U,\tau) = \frac{1}{2} u_{-1}^2 \operatorname{tr}\left( \gamma U_{12}^\top \mu_\tau \mu_\tau^\top U_{12} \Lambda + U_{22}^\top \Gamma \left( \Lambda + \mu_\tau \mu_\tau^\top \right) U_{22} \Lambda + 2 U_{12}^\top \mu_\tau \mu_\tau^\top \left( \Gamma - \frac{2}{n} \mu_\tau \mu_\tau^\top \right) U_{22} \Lambda \right)$$
$$- u_{-1} \operatorname{tr}\left( \left( \mu_\tau \mu_\tau^\top U_{12} \Lambda + \left( \Lambda + \mu_\tau \mu_\tau^\top \right) U_{22} \Lambda \right) \right) + \frac{1}{2} \operatorname{tr}(\Lambda) \tag{40}$$
$$:= -c_{1,1} u_{-1} + c_{1,2} u_{-1}^2 + \frac{1}{2} \operatorname{tr}(\Lambda)$$

where $-c_{1,1}$ is the coefficient of 1st degree term $u_{-1}$ and $c_{1,2}$ is the coefficient of 2nd degree term $u_{-1}^2$ in $\ell_1$.

Similarly we have

$$\ell_2(U,\tau) = \frac{1}{2} u_{-1}^2 \operatorname{tr}\left( \gamma (U_{12} + U_{11})^\top \mu_\tau \mu_\tau^\top (U_{12} + U_{11}) + (U_{22} + U_{21})^\top \Gamma \left( \Lambda + \mu_\tau \mu_\tau^\top \right) (U_{22} + U_{21}) \mu_\tau \mu_\tau^\top \right.$$
$$\left. + 2 (U_{12} + U_{11})^\top \mu_\tau \mu_\tau^\top \left( \Gamma - \frac{2}{n} \mu_\tau \mu_\tau^\top \right) (U_{22} + U_{21}) \mu_\tau \mu_\tau^\top \right)$$
$$- u_{-1} \operatorname{tr}\left( \mu_\tau \mu_\tau^\top (U_{12} + U_{11}) \mu_\tau \mu_\tau^\top + \left( \Lambda + \mu_\tau \mu_\tau^\top \right) (U_{22} + U_{21}) \mu_\tau \mu_\tau^\top \right) + \frac{1}{2} \operatorname{tr}\left( \mu_\tau \mu_\tau^\top \right) \tag{41}$$
$$:= -c_{2,1} u_{-1} + c_{2,2} u_{-1}^2 + \frac{1}{2} \operatorname{tr}\left( \mu_\tau \mu_\tau^\top \right)$$

where $-c_{2,1}$ is the coefficient of 1st degree term $u_{-1}$ and $c_{2,2}$ is the coefficient of 2nd degree term $u_{-1}^2$ in $\ell_2$.

Then we have

$$L(U) = \mathbb{E}_{\mu_\tau} \left[ \ell_1(U,\tau) + \ell_2(U,\tau) \right]$$
$$= \mathbb{E}_{\mu_\tau} \left[ (c_{1,2} + c_{2,2}) u_{-1}^2 - (c_{2,1} + c_{1,1}) u_{-1} \right] + \frac{1}{2} \mathbb{E}_{\mu_\tau} \left[ \|\mu_\tau\|^2 \right] + \frac{1}{2} \operatorname{tr}(\Lambda) \tag{42}$$

Now we give several useful lower and upper bounds on $c_{1,2} + c_{2,2}$ and $c_{2,1} + c_{1,1}$. Denote $c_{i,j}(t)$ as the corresponding coefficient at time $t$ under the gradient flow. We have the following claim.

**Claim 1.** *We have the following three bounds:*

*1.*

$$\mathbb{E}[c_{1,1}(0) + c_{2,1}(0)] \geq (d+2)u_{-1}(0), \tag{43}$$

*2.*

$$c_{1,2} + c_{2,2} \leq u_{-1}^2 \|\Lambda + \mu_\tau \mu_\tau^\top\|_F^2 \left(2\|\mu_\tau\|^2 + \|\Lambda\|_F\right), \tag{44}$$

*3.*

$$c_{1,1} + c_{2,1} \leq (2 + \sqrt{2})u_{-1}\|\Lambda + \mu_\tau \mu_\tau^\top\|_F^2. \tag{45}$$

Now we can upper bound $L(U(0))$.

$$L(U(0)) = \mathbb{E}\left[\ell_1\left(U(0), \tau\right) + \ell_2\left(U(0), \tau\right)\right]$$

$$= \mathbb{E}\left[(c_{1,2} + c_{2,2})u_{-1}(0)^2 - (c_{2,1} + c_{1,1})u_{-1}(0)\right] + \frac{1}{2}\mathbb{E}\left[\|\mu_\tau\|^2\right] + \frac{1}{2}\operatorname{tr}(\Lambda)$$

$$\leq u_{-1}^2(0)\mathbb{E}\left[\|\Lambda + \mu_\tau\mu_\tau^\top\|_F^2\left(\|\Lambda\|_F + 2\|\mu_\tau\|^2\right)u_{-1}^2(0) - (d+2)\right] + \frac{1}{2}\mathbb{E}\left[\|\mu_\tau\|^2\right] + \frac{1}{2}\operatorname{tr}(\Lambda) \quad \text{((43) and (44))}$$

$$\leq -\frac{1}{2}(d+2)u_{-1}^2(0) + \frac{1}{2}\mathbb{E}\left[\|\mu_\tau\|^2\right] + \frac{1}{2}\operatorname{tr}(\Lambda) \tag{46}$$

The last inequality comes from that $u_{-1}(0) < \alpha = \left(\frac{d+2}{2\,\mathbb{E}\left[\|\Lambda + \mu_\tau\mu_\tau^\top\|_F^2\left(\|\Lambda\|_F + 2\|\mu_\tau\|^2\right)\right]}\right)^{\frac{1}{2}}$.

Note that when $u_{-1} = 0$, the loss $L(U) = \frac{1}{2}\mathbb{E}_{\mu_\tau}\left[\|\mu_\tau\|^2\right] + \frac{1}{2}\operatorname{tr}(\Lambda)$. Therefore, $u_{-1}$ is non-zero whenever $L(U) < \frac{1}{2}\mathbb{E}_{\mu_\tau}\left[\|\mu_\tau\|^2\right] + \frac{1}{2}\operatorname{tr}(\Lambda)$. Since $L(U)$ is non-increasing and by (46) we know $L(U(0)) < \frac{1}{2}\mathbb{E}_{\mu_\tau}\left[\|\mu_\tau\|^2\right] + \frac{1}{2}\operatorname{tr}(\Lambda)$, we have $L(U) < \frac{1}{2}\mathbb{E}_{\mu_\tau}\left[\|\mu_\tau\|^2\right] + \frac{1}{2}\operatorname{tr}(\Lambda)$ for all $t \geq 0$, which implies that $u_{-1}$ is non-zero for all $t \geq 0$. Further since we have $u_{-1}(0) > 0$ and $u_{-1}(t)$ is continuous on $t$, we have $u_{-1} > 0$ for all $t \geq 0$.

Now we lower bound $L(U)$.

$$L(U) = \mathbb{E}\left[(c_{1,2} + c_{2,2})u_{-1}^2 - (c_{2,1} + c_{1,1})u_{-1}\right] + \frac{1}{2}\mathbb{E}\left[\|\mu_\tau\|^2\right] + \frac{1}{2}\operatorname{tr}(\Lambda)$$

$$\geq -u_{-1}\mathbb{E}\left[c_{2,1} + c_{1,1}\right] + \frac{1}{2}\mathbb{E}\left[\|\mu_\tau\|^2\right] + \frac{1}{2}\operatorname{tr}(\Lambda) \tag{47}$$

$$\geq -u_{-1}^2\mathbb{E}\left[(2 + \sqrt{2})\|\Lambda + \mu_\tau\mu_\tau^\top\|_F^2\right] + \frac{1}{2}\mathbb{E}\left[\|\mu_\tau\|^2\right] + \frac{1}{2}\operatorname{tr}(\Lambda) \quad (u_{-1} > 0 \text{ and (45)})$$

Since $L(U) \leq L(U(0))$, combining (46) and (47) we have

$$u_{-1} \geq \frac{(d+2)u_{-1}(0)}{(4 + 2\sqrt{2})\mathbb{E}\left[\|\Lambda + \mu_\tau\mu_\tau^\top\|_F^2\right]} = \frac{(d+2)u_{-1}(0)}{(4 + 2\sqrt{2})\left(\|\Lambda\|_F^2 + 2\operatorname{tr}(\Lambda) + d^2 + 2d\right)} = \beta > 0. \tag{48}$$

It remains to prove Claim 1.

*Proof of (43).* Recall that

$$c_{1,1} = \operatorname{tr}\left(\left(\mu_\tau\mu_\tau^\top U_{12}\Lambda + \left(\Lambda + \mu_\tau\mu_\tau^\top\right)U_{22}\Lambda\right)\right) \tag{49}$$

and

$$c_{2,1} = \operatorname{tr}\left(\mu_\tau\mu_\tau^\top\left(U_{12} + U_{11}\right)\mu_\tau\mu_\tau^\top + \left(\Lambda + \mu_\tau\mu_\tau^\top\right)\left(U_{22} + U_{21}\right)\mu_\tau\mu_\tau^\top\right). \tag{50}$$

Computing the expectation, we have

$$\mathbb{E}[c_{1,1}] = \operatorname{tr}\left(U_{12}\Lambda + U_{22}\Lambda + U_{22}\Lambda^2\right) \tag{51}$$

and

$$\mathbb{E}[c_{2,1}] = (d+2)\operatorname{tr}(U_{12} + U_{11} + U_{22} + U_{21}) + \operatorname{tr}((U_{22} + U_{21})\Lambda). \tag{52}$$

By Assumption 2.1, at time $t = 0$ we have $U_{12}(0), U_{11}(0), U_{22}(0)$ and $U_{21}(0)$ are PSD matrices. Therefore we have $\mathbb{E}[c_{1,1}(0)] \geq 0$ and $\mathbb{E}[c_{2,1}(0)] \geq (d+2)\operatorname{tr}(U_{12}(0) + U_{11}(0) + U_{22}(0) + U_{21}(0))$, which implies that

$$
\begin{aligned}
\mathbb{E}[c_{1,1}(0) + c_{2,1}(0)]^2 &\geq (d+2)^2 \left( \|\sqrt{U_{12}(0)}\|_F^2 + \|\sqrt{U_{11}(0)}\|_F^2 + \|\sqrt{U_{22}(0)}\|_F^2 + \|\sqrt{U_{21}(0)}\|_F^2 \right)^2 \\
&\geq (d+2)^2 \left( \|\sqrt{U_{12}(0)}\|_F^4 + \|\sqrt{U_{11}(0)}\|_F^4 + \|\sqrt{U_{22}(0)}\|_F^4 + \|\sqrt{U_{21}(0)}\|_F^4 \right) \\
&\geq (d+2)^2 \left( \|U_{12}(0)\|_F^2 + \|U_{11}(0)\|_F^2 + \|U_{22}(0)\|_F^2 + \|U_{21}(0)\|_F^2 \right) \quad \text{(submultiplicativity)} \\
&= (d+2)^2 u_{-1}(0)^2 \quad \text{(Assumption 2.1)}
\end{aligned}
\tag{53}
$$

Therefore we have $\mathbb{E}[c_{1,1}(0) + c_{2,1}(0)] \geq (d+2)u_{-1}(0)$. $\qquad\square$

*Proof of (44).* Recall that

$$c_{1,2} = \frac{1}{2}\operatorname{tr}\left( \gamma U_{12}^\top \mu_\tau \mu_\tau^\top U_{12}\Lambda + U_{22}^\top \Gamma \left(\Lambda + \mu_\tau \mu_\tau^\top\right) U_{22}\Lambda + 2U_{12}^\top \mu_\tau \mu_\tau^\top \left(\Gamma - \frac{2}{n}\mu_\tau \mu_\tau^\top\right) U_{22}\Lambda \right) \tag{54}$$

and

$$
\begin{aligned}
c_{2,2} = \frac{1}{2}\operatorname{tr}\Big( & \gamma \left(U_{12} + U_{11}\right)^\top \mu_\tau \mu_\tau^\top \left(U_{12} + U_{11}\right) \mu_\tau \mu_\tau^\top + \left(U_{22} + U_{21}\right)^\top \Gamma \left(\Lambda + \mu_\tau \mu_\tau^\top\right) \left(U_{22} + U_{21}\right) \mu_\tau \mu_\tau^\top \\
& + 2\left(U_{12} + U_{11}\right)^\top \mu_\tau \mu_\tau^\top \left(\Gamma - \frac{2}{n}\mu_\tau \mu_\tau^\top\right) \left(U_{22} + U_{21}\right) \mu_\tau \mu_\tau^\top \Big).
\end{aligned}
\tag{55}
$$

Note that $\Gamma \to \Lambda + \mu_\tau \mu_\tau^\top$ and $\gamma \to \|\mu_\tau\|^2$ if $n \to \infty$. Therefore we have

$$
\begin{aligned}
c_{2,2} =& \frac{1}{2}\operatorname{tr}\Big( \|\mu_\tau\|^2 \left(U_{12} + U_{11}\right)^\top \mu_\tau \mu_\tau^\top \left(U_{12} + U_{11}\right) \mu_\tau \mu_\tau^\top + \left(U_{22} + U_{21}\right)^\top \left(\Lambda + \mu_\tau \mu_\tau^\top\right)^2 \left(U_{22} + U_{21}\right) \mu_\tau \mu_\tau^\top \\
& \quad + 2\left(U_{12} + U_{11}\right)^\top \mu_\tau \mu_\tau^\top \left(\Lambda + \mu_\tau \mu_\tau^\top\right) \left(U_{22} + U_{21}\right) \mu_\tau \mu_\tau^\top \Big) \\
\leq& \frac{1}{2}\|\mu_\tau\|^2 \|U_{12} + U_{11}\|_F^2 \|\mu_\tau \mu_\tau^\top\|_F^2 + \frac{1}{2}\|U_{22} + U_{21}\|_F^2 \|\Lambda + \mu_\tau \mu_\tau^\top\|_F^2 \|\mu_\tau \mu_\tau^\top\|_F \\
& \quad + \|U_{12} + U_{11}\|_F \|U_{22} + U_{21}\|_F \|\mu_\tau \mu_\tau^\top\|_F^2 \|\Lambda + \mu_\tau \mu_\tau^\top\|_F \quad \text{(Cauchy-Schwartz inequality)} \\
\leq& \|\Lambda + \mu_\tau \mu_\tau^\top\|_F^2 \|\mu_\tau \mu_\tau^\top\|_F \left( \frac{1}{2}\|U_{12} + U_{11}\|_F^2 + \frac{1}{2}\|U_{22} + U_{21}\|_F^2 + \|U_{12} + U_{11}\|_F \|U_{22} + U_{21}\|_F \right) \\
=& \frac{1}{2}\|\Lambda + \mu_\tau \mu_\tau^\top\|_F^2 \|\mu_\tau \mu_\tau^\top\|_F \left( \|U_{12} + U_{11}\|_F + \|U_{22} + U_{21}\|_F \right)^2 \\
\leq& 2\|\Lambda + \mu_\tau \mu_\tau^\top\|_F^2 \|\mu_\tau \mu_\tau^\top\|_F \left( \|U_{12}\|_F^2 + \|U_{11}\|_F^2 + \|U_{22}\|_F^2 + \|U_{21}\|_F^2 \right) \quad \text{(Triangle inequality)} \\
=& 2\|\Lambda + \mu_\tau \mu_\tau^\top\|_F^2 \|\mu_\tau\|^2 u_{-1}^2 \quad \text{(Lemma A.1)}
\end{aligned}
\tag{56}
$$

Here the second last inequality comes from $\|\mu_\tau\|^2 = \|\mu_\tau \mu_\tau^\top\|_F \leq \|\Lambda + \mu_\tau \mu_\tau^\top\|_F$.

Similarly, for $c_{1,2}$ we have

$$
\begin{aligned}
c_{1,2} =& \frac{1}{2} \operatorname{tr}\left(\|\mu_\tau\|^2 U_{12}^\top \mu_\tau \mu_\tau^\top U_{12}\Lambda + U_{22}^\top \left(\Lambda + \mu_\tau \mu_\tau^\top\right)^2 U_{22}\Lambda + 2U_{12}^\top \mu_\tau \mu_\tau^\top \left(\Lambda + \mu_\tau \mu_\tau^\top\right) U_{22}\Lambda\right) \\
\leq& \frac{1}{2}\|\mu_\tau\|^2 \|U_{12}\|_F^2 \|\Lambda\|_F^2 + \frac{1}{2}\|U_{22}\|_F^2\|\Lambda + \mu_\tau \mu_\tau^\top\|_F^2 \|\Lambda\|_F + \|\mu_\tau \mu_\tau^\top\|_F^2\|\Lambda\|_F\|U_{11}\|_F\|U_{12}\|_F \quad \text{(Cauchy-Schwartz inequality)} \\
\leq& \left\|\Lambda + \mu_\tau \mu_\tau^\top\right\|_F^2 \|\Lambda\|_F \left(\frac{1}{2}\|U_{12}\|_F^2 + \frac{1}{2}\|U_{22}\|_F^2 + \|U_{12}\|_F \|U_{22}\|_F\right) \\
=& \frac{1}{2}\left\|\Lambda + \mu_\tau \mu_\tau^\top\right\|_F^2 \|\Lambda\|_F \left(\|U_{12}\|_F + \|U_{22}\|_F\right)^2 \\
\leq& \frac{1}{2}\left\|\Lambda + \mu_\tau \mu_\tau^\top\right\|_F^2 \|\Lambda\|_F \left(\|U_{12}\|_F + \|U_{22}\|_F + \|U_{11}\|_F^2 + \|U_{21}\|_F^2\right)^2 \\
\leq& \left\|\Lambda + \mu_\tau \mu_\tau^\top\right\|_F^2 \|\Lambda\|_F \left(\|U_{12}\|_F^2 + \|U_{11}\|_F^2 + \|U_{22}\|_F^2 + \|U_{21}\|_F^2\right) \quad \text{(Cauchy-Schwartz inequality)} \\
=& \left\|\Lambda + \mu_\tau \mu_\tau^\top\right\|_F^2 \|\Lambda\|_F \, u_{-1}^2 \quad \text{(Lemma A.1)}
\end{aligned}
\tag{57}
$$

Here the second inequality comes from $\|\mu_\tau\|^2 \leq \|\Lambda + \mu_\tau \mu_\tau^\top\|_F$ and $\|\Lambda\|_F \leq \|\Lambda + \mu_\tau \mu_\tau^\top\|_F$.

Adding (57) and (56) up, we have

$$
c_{1,2} + c_{2,2} \leq u_{-1}^2 \|\Lambda + \mu_\tau \mu_\tau^\top\|_F^2 \left(2\|\mu_\tau\|^2 + \|\Lambda\|_F\right).
\tag{58}
$$

$\square$

*Proof of (45).* Recall that

$$
c_{1,1} = \operatorname{tr}\left(\left(\mu_\tau \mu_\tau^\top U_{12}\Lambda + \left(\Lambda + \mu_\tau \mu_\tau^\top\right) U_{22}\Lambda\right)\right)
\tag{59}
$$

and

$$
c_{2,1} = \operatorname{tr}\left(\mu_\tau \mu_\tau^\top \left(U_{12} + U_{11}\right) \mu_\tau \mu_\tau^\top + \left(\Lambda + \mu_\tau \mu_\tau^\top\right) \left(U_{22} + U_{21}\right) \mu_\tau \mu_\tau^\top\right).
\tag{60}
$$

We have

$$
\begin{aligned}
c_{1,1} =& \operatorname{tr}\left(\left(\mu_\tau \mu_\tau^\top U_{12}\Lambda + \left(\Lambda + \mu_\tau \mu_\tau^\top\right) U_{22}\Lambda\right)\right) \\
\leq& \|\mu_\tau \mu_\tau^\top\|_F \|\Lambda\|_F \|U_{12}\|_F + \|\Lambda + \mu_\tau \mu_\tau^\top\|_F \|\Lambda\|_F \|U_{22}\|_F \quad \text{(Cauchy-Schwartz inequality)} \\
\leq& \left(\|U_{12}\|_F + \|U_{22}\|_F\right) \left\|\Lambda + \mu_\tau \mu_\tau^\top\right\|_F^2 \\
\leq& \sqrt{2\left(\|U_{12}\|_F^2 + \|U_{22}\|_F^2\right)} \left\|\Lambda + \mu_\tau \mu_\tau^\top\right\|_F^2 \quad \text{(Cauchy-Schwartz inequality)} \\
\leq& \sqrt{2\left(\|U_{12}\|_F^2 + \|U_{22}\|_F^2 + \|U_{11}\|_F^2 + \|U_{21}\|_F^2\right)} \left\|\Lambda + \mu_\tau \mu_\tau^\top\right\|_F^2 \\
=& \sqrt{2}u_{-1} \left\|\Lambda + \mu_\tau \mu_\tau^\top\right\|_F^2 \quad \text{(Lemma A.1)}.
\end{aligned}
\tag{61}
$$

Here the second inequality comes from $\|\mu_\tau \mu_\tau^\top\|_F \leq \|\Lambda + \mu_\tau \mu_\tau^\top\|_F$ and $\|\Lambda\|_F \leq \|\Lambda + \mu_\tau \mu_\tau^\top\|_F$.

Similarly we have

$$
\begin{aligned}
c_{2,1} =& \operatorname{tr}\left(\mu_\tau \mu_\tau^\top \left(U_{12} + U_{11}\right) \mu_\tau \mu_\tau^\top + \left(\Lambda + \mu_\tau \mu_\tau^\top\right) \left(U_{22} + U_{21}\right) \mu_\tau \mu_\tau^\top\right) \\
\leq& \|\mu_\tau \mu_\tau^\top\|_F^2 \|U_{12} + U_{11}\|_F + \|\Lambda + \mu_\tau \mu_\tau^\top\|_F \|\mu_\tau \mu_\tau^\top\|_F \|U_{22} + U_{21}\|_F \quad \text{(Cauchy-Schwartz inequality)} \\
\leq& \left(\|U_{12} + U_{11}\|_F + \|U_{22} + U_{21}\|_F\right) \left\|\Lambda + \mu_\tau \mu_\tau^\top\right\|_F^2 \\
\leq& \left(\|U_{12}\|_F + \|U_{11}\|_F + \|U_{22}\|_F + U_{21}\|_F\right) \left\|\Lambda + \mu_\tau \mu_\tau^\top\right\|_F^2 \quad \text{(Triangle inequality)} \\
\leq& 2\sqrt{\|U_{12}\|_F^2 + \|U_{22}\|_F^2 + \|U_{11}\|_F^2 + \|U_{21}\|_F^2} \left\|\Lambda + \mu_\tau \mu_\tau^\top\right\|_F^2 \quad \text{(Cauchy-Schwartz inequality)} \\
=& 2u_{-1} \left\|\Lambda + \mu_\tau \mu_\tau^\top\right\|_F^2. \quad \text{(Lemma A.1)}
\end{aligned}
\tag{62}
$$

Here the second inequality comes from $\|\mu_\tau \mu_\tau^\top\|_F \leq \|\Lambda + \mu_\tau \mu_\tau^\top\|_F$.

Adding (61) and (62) up, we have

$$c_{1,1} + c_{2,1} \leq (2 + \sqrt{2})u_{-1}\|\Lambda + \mu_\tau\mu_\tau^\top\|_F^2. \tag{63}$$

□

□

*Proof of Lemma A.3.* We take a new parameterization $\widetilde{U}_{12} := u_{-1}U_{12}$ and $\widetilde{U}_{22} := u_{-1}U_{22}$. Denote $L_1(U) = \mathbb{E}[\ell_1(U,\tau)]$ and $L_2(U) = \mathbb{E}[\ell_2(U,\tau)]$. Then we can simply write $L_1(U)$ in the new parameterization $\widetilde{U} = (\widetilde{U}_{12}, \widetilde{U}_{22})$ as $L_1(\widetilde{U})$. Specifically, we have

$$
\begin{aligned}
L_1(U) =& L_1(\widetilde{U}) \\
=& \frac{1}{2}\mathbb{E}\left[\mathrm{tr}\left(\|\mu_\tau\|^2\widetilde{U}_{12}^\top\mu_\tau\mu_\tau^\top\widetilde{U}_{12}\Lambda + \widetilde{U}_{22}^\top\left(\Lambda + \mu_\tau\mu_\tau^\top\right)\widetilde{U}_{22}\Lambda + \Lambda \right.\right. \\
&\left.\left. + 2\widetilde{U}_{12}^\top\mu_\tau\mu_\tau^\top\left(\Lambda + \mu_\tau\mu_\tau^\top\right)\widetilde{U}_{22}\Lambda - 2\mu_\tau\mu_\tau^\top\widetilde{U}_{12}\Lambda - 2\left(\Lambda + \mu_\tau\mu_\tau^\top\right)\widetilde{U}_{22}\Lambda\right)\right].
\end{aligned}
\tag{64}
$$

First we want to show that

$$\left\|\nabla L_1(\widetilde{U})\right\|_F \geq \frac{1}{10}\lambda_{\min}(\Lambda)\min\{\lambda_{\min}(\Lambda)^3, 1\}\|\widetilde{U} - \widetilde{U}_*\|_F \tag{65}$$

for all $t \geq 0$, where $\widetilde{U}_* := (-\Lambda^{-1}, \Lambda^{-1})$.

By direct computation, we have the gradients

- $\frac{\partial L_1(\widetilde{U})}{\partial \widetilde{U}_{12}} = \left((d+2)\widetilde{U}_{12} + (d+2)\widetilde{U}_{22} + \Lambda\widetilde{U}_{22} - I\right)\Lambda;$

- $\frac{\partial L_1(\widetilde{U})}{\partial \widetilde{U}_{22}} = \left((d+2)\widetilde{U}_{22} + \left(\Lambda^2 + 2\Lambda\right)\widetilde{U}_{22} + (d+2)\widetilde{U}_{12} + \Lambda\widetilde{U}_{12} - I - \Lambda\right)\Lambda.$

Therefore we have

$$
\|\nabla L_1(\widetilde{U})\|_F \cdot \|\widetilde{U} - \widetilde{U}_*\|_F
$$

$$
\geq \left\langle \left( \frac{\partial L_1(\widetilde{U})}{\partial \widetilde{U}_{12}}, \frac{\partial L_1(\widetilde{U})}{\partial \widetilde{U}_{22}} \right), \left( \widetilde{U}_{12} + \Lambda^{-1}, \widetilde{U}_{22} - \Lambda^{-1} \right) \right\rangle
$$

$$
= \operatorname{tr} \left( d \left( \widetilde{U}_{12} + \widetilde{U}_{22} \right) \Lambda \left( \widetilde{U}_{12}^\top + \widetilde{U}_{22}^\top \right) + \Lambda \left[ 2 \left( \widetilde{U}_{22} + \widetilde{U}_{22} \right)^\top \left( \widetilde{U}_{12} + \widetilde{U}_{22} \right) \right. \right.
$$

$$
+ \left( \widetilde{U}_{22} - \Lambda^{-1} \right) \Lambda \left( \widetilde{U}_{12}^\top + \Lambda^{-1} \right) + (\Lambda + 2I) \left( \widetilde{U}_{22} - \Lambda^{-1} \right) \Lambda \left( \widetilde{U}_{22}^\top - \Lambda^{-1} \right)
$$

$$
\left. \left. + \left( \widetilde{U}_{12} + \Lambda^{-1} \right) \Lambda \left( \widetilde{U}_{22}^\top - \Lambda^{-1} \right) \right] \right)
$$

$$
\geq \operatorname{tr} \left( \Lambda \left( 2 \left( \widetilde{U}_{12} + \widetilde{U}_{22} \right)^\top \left( \widetilde{U}_{12} + \widetilde{U}_{22} \right) + \left( \widetilde{U}_{22} - \Lambda^{-1} \right) \Lambda \left( \widetilde{U}_{12}^\top + \Lambda^{-1} \right) \right. \right.
$$

$$
\left. \left. + (\Lambda + 2I) \left( \widetilde{U}_{22} - \Lambda^{-1} \right) \Lambda \left( \widetilde{U}_{22}^\top - \Lambda^{-1} \right) + \left( \widetilde{U}_{12} + \Lambda^{-1} \right) \Lambda \left( \widetilde{U}_{22}^\top - \Lambda^{-1} \right) \right) \right)
$$

$$
= \operatorname{tr} \left( \Lambda \left( 2 \left( \widetilde{U}_{12} + \Lambda^{-1} \right)^\top \left( \widetilde{U}_{12} + \Lambda^{-1} \right) + 2 \left( \widetilde{U}_{22} - \Lambda^{-1} \right)^\top \left( \widetilde{U}_{22} - \Lambda^{-1} \right) \right. \right.
$$

$$
+ 4 \left( \widetilde{U}_{12} + \Lambda^{-1} \right) \left( \widetilde{U}_{22} - \Lambda^{-1} \right) + \left( \widetilde{U}_{12} + \Lambda^{-1} \right)^\top \Lambda \left( \widetilde{U}_{22} - \Lambda^{-1} \right)
$$

$$
\left. \left. + \left( \widetilde{U}_{22}^\top - \Lambda^{-1} \right) \Lambda (\Lambda + 2I) \left( \widetilde{U}_{22} - \Lambda^{-1} \right) + \left( \widetilde{U}_{12}^\top + \Lambda^{-1} \right) \Lambda \left( \widetilde{U}_{22} - \Lambda^{-1} \right) \right) \right)
$$

$$
= \operatorname{tr} \left( \Lambda \left( 2 \left( \widetilde{U}_2 + \Lambda^{-1} \right)^\top \left( \widetilde{U}_2 + \Lambda^{-1} \right) + \left( \widetilde{U}_{12} + \Lambda^{-1} \right)^\top (2\Lambda + 4I) \left( \widetilde{U}_{22} - \Lambda^{-1} \right) + \left( \widetilde{U}_{22} - \Lambda^{-1} \right) \left( \Lambda^\top + 2\Lambda + 2I \right) \left( \widetilde{U}_{22} - \Lambda^{-1} \right) \right) \right)
$$

$$
= \operatorname{tr} \left( \Lambda \left( VV^\top + \left( \widetilde{U}_{12} + \Lambda^{-1} \right) \underbrace{ \frac{1}{2} \Lambda^2 \left( \Lambda^2 + 2\Lambda + 2I \right)^{-1} }_{P_1} \left( \widetilde{U}_{12} + \Lambda^{-1} \right)^\top \right. \right.
$$

$$
\left. \left. + \left( \widetilde{U}_{22} - \Lambda^{-1} \right) \underbrace{ \left( \Lambda^2 + 2\Lambda + 2I - (\Lambda + 2I)^2 \left( 2I - \frac{1}{2} \Lambda^2 \left( \Lambda^2 + 2\Lambda + 2I \right)^{-1} \right) \right) }_{P_2} \right) \right).
$$

$$\tag{66}$$

In the last equation the matrix $V$ is defined as

$$
V := \left( \widetilde{U}_2 + \Lambda^{-1} \right) \left( 2I - \frac{1}{2} \Lambda^2 \left( \Lambda^2 + 2\Lambda + 2I \right)^{-1} \right)^{\frac{1}{2}} + \left( \widetilde{U}_{22} - \Lambda^{-1} \right) (\Lambda + 2I) \left( 2I - \frac{1}{2} \Lambda^2 \left( \Lambda^2 + 2 + 2I \right)^{-1} \right)^{-\frac{1}{2}}.
$$

$$\tag{67}$$

It is easy to see $P_1$ is a diagonal PSD matrix and $P_2$ is a diagonal matrix. Actually $P_2$ is also PSD. To see this, for any diagonal entry $a$ in $\Lambda$, the corresponding diagonal entry in $P_2$ is $a^2 + 2a + 2 - \frac{2(a^2 + 2a + 4)(a^2 + 2a + 2)}{3a^2 + 8a + 8} \geq \frac{1}{4} a^2 \geq 0$. Furthermore we have $\lambda_{\min}(P_2) \geq \frac{1}{4} \lambda_{\min}(\Lambda)^2$. Similarly, we have $\lambda_{\min}(P_1) \geq \frac{1}{10} \min\{\lambda_{\min}(\Lambda)^3, 1\}$. Removing the term containing $V$ in (66), we have

$$
\|\nabla L_1(\widetilde{U})\|_F \cdot \|\widetilde{U} - \widetilde{U}_*\|_F
$$

$$
\geq \operatorname{tr} \left( \Lambda \left( \widetilde{U}_{12} + \Lambda^{-1} \right) P_1 \left( \widetilde{U}_{12} + \Lambda^{-1} \right)^\top + \Lambda \left( \widetilde{U}_{22} - \Lambda^{-1} \right) P_2 \left( \widetilde{U}_{22} - \Lambda^{-1} \right)^\top \right)
$$

$$
\geq \frac{1}{10} \lambda_{\min}(\Lambda) \min\{\lambda_{\min}(\Lambda)^3, 1\} \|\widetilde{U}_{12} + \Lambda^{-1}\|_F^2 + \frac{1}{4} \lambda_{\min}(\Lambda)^2 \|\widetilde{U}_{22} - \Lambda^{-1}\|_F^2
$$

$$
\geq \frac{1}{10} \lambda_{\min}(\Lambda) \min\{\lambda_{\min}(\Lambda)^3, 1\} \left( \|\widetilde{U}_{12} + \Lambda^{-1}\|_F^2 + \|\widetilde{U}_{22} - \Lambda^{-1}\|_F^2 \right)
$$

$$
= \frac{1}{10} \lambda_{\min}(\Lambda) \min\{\lambda_{\min}(\Lambda)^3, 1\} \|\widetilde{U} - \widetilde{U}_*\|_F^2.
$$

$$\tag{68}$$

Therefore we have

$$\|\nabla L_1(\widetilde{U})\|_F \geq \frac{1}{10} \lambda_{\min}(\Lambda) \min\{\lambda_{\min}(\Lambda)^3, 1\} \|\widetilde{U} - \widetilde{U}_*\|_F. \tag{69}$$

Now we derive a gradient lower bound for $L_2$. Define $\bar{A} := \frac{1}{2}(A + A^\top)$ for any $d \times d$ matrix $A$. We define a new parameterization $U_1 := u_{-1}(\bar{U}_{11} + \bar{U}_{12} + \bar{U}_{21} + \bar{U}_{22})$ and $U_2 := u_{-1}(U_{21} + U_{22})$. By checking the dynamics of $U_{11}$ and $U_{12}$ in (25) and (27), we can find that $U_{11}$ and $U_{12}$ keep symmetric for all $t \geq 0$. Therefore $U_1 := u_{-1}(U_{11} + U_{12} + \bar{U}_{21} + \bar{U}_{22})$. Note that for any $d \times d$ matrix $A$, it holds that $\mu_\tau^\top A \mu_\tau = \mu_\tau^\top \bar{A} \mu_\tau$. Therefore we can write $L_2$ under the new parameterization as $L_2(U_1, U_2)$:

$$L_2(U) = L_2(U_1, U_2) = \frac{1}{2}\mathbb{E}\left[\left\|\left(\Lambda U_2 - I + \mu_\tau \mu_\tau^\top U_1\right)\mu_\tau\right\|^2\right]. \tag{70}$$

Therefore we know $L_2$ is convex in terms of $U_1$ and $U_2$ since $\Lambda U_2 - I + \mu_\tau \mu_\tau^\top U_1$ is an affine function of $(U_1, U_2)$, $f(X) = \|X\mu_\tau\|^2$ is a convex function and the expectation reserves convexity. By setting $U_2^* = \Lambda^{-1}$ and $U_1^* = 0_{d \times d}$ in (70), we have $L_2(U_1^*, U_2^*) = 0$. Denote $\hat{U} = (U_1, U_2)$, $\hat{U}_* = (U_1^*, U_2^*)$.

By convexity, we have

$$\left\langle \nabla L_2(\hat{U}), \hat{U} - \hat{U}_* \right\rangle + L_2(U_1^*, U_2^*) = \left\langle \nabla L_2(\hat{U}), \hat{U} - \hat{U}_* \right\rangle \geq L_2(\hat{U}). \tag{71}$$

Therefore expanding the expectation in (70), we have

$$\begin{aligned}
L_2(\hat{U}) &= \|\Lambda U_2 - I\|_F^2 + (d+4)\left(\text{tr}\left(U_1\right)^2 + \text{tr}\left(U_1^2\right) + \|U_1\|_F^2\right) \\
&\quad + 2\left(\text{tr}\left((\Lambda U_2 - I)(U_1)\right) + \text{tr}\left((\Lambda U_2 - I)(U_1)^\top\right) + \text{tr}\left(\Lambda U_2 - I\right)\text{tr}\left(U_1\right)\right) \\
&= \left\|\frac{1}{\sqrt{d+4}}\left(\Lambda U_2 + U_2^\top \Lambda - 2I\right) + \sqrt{d+4}\left(U_1\right)\right\|_F^2 + \frac{d}{d+4}\|\Lambda U_2 - I\|_F^2 + (d+4)\|U_1\|_F^2 \\
&\quad + (d+4)\text{tr}\left(U_1\right)^2 + 2\text{tr}\left(\Lambda U_2 - I\right)\text{tr}\left(U_1\right) \\
&\geq \left\|\frac{1}{\sqrt{d+4}}\left(\Lambda U_2 + U_2^\top \Lambda - 2I\right) + \sqrt{d+4}\left(U_1\right)\right\|_F^2 + \left(\frac{d}{d+4} - \frac{d}{d+5}\right)\|\Lambda U_2 - I\|_F^2 + \frac{1}{d+5}\text{tr}\left(\Lambda U_2 - I\right)^2 \\
&\quad + 4\|U_1\|_F^2 + (d+5)\text{tr}\left(U_1\right)^2 + 2\text{tr}\left(\Lambda U_2 - I\right)\text{tr}\left(U_1\right) \\
&= \left\|\frac{1}{\sqrt{d+4}}\left(\Lambda U_2 + U_2^\top \Lambda - 2I\right) + \sqrt{d+4}\left(U_1\right)\right\|_F^2 + \left(\frac{d}{d+4} - \frac{d}{d+5}\right)\|\Lambda U_2 - I\|_F^2 + 4\|U_1\|_F^2 \\
&\quad + \left(\frac{1}{\sqrt{d+5}}\text{tr}\left(U_2 - I\right) + \sqrt{d+5}\,\text{tr}\left(U_1\right)\right)^2 \\
&\geq \left(\frac{d}{d+4} - \frac{d}{d+5}\right)\|\Lambda U_2 - I\|_F^2 + 4\|U_1\|_F^2 \\
&\geq \frac{1}{30d}\left(\|\Lambda U_2 - I\|_F^2 + \|U_1\|_F^2\right) \\
&\geq \frac{\min\left(\lambda_{\min}(\Lambda), 1\right)^2}{30d}\left(\|U_1\|_F^2 + \|U_2 - \Lambda^{-1}\|_F^2\right) \\
&= \frac{\min\left(\lambda_{\min}(\Lambda), 1\right)^2}{30d}\|\hat{U} - \hat{U}_*\|_F^2.
\end{aligned} \tag{72}$$

Here the second equation comes from that $U_1$ is symmetric hence $\text{tr}\left(U_1^2\right) = \|U_1\|_F^2$. The first inequality comes from that $\|A\|_F^2 \geq \frac{1}{d}\text{tr}(A)^2$ for any $d \times d$ real matrix $A$. The last inequality comes from that

$$\left\|\begin{pmatrix} U_2 - \Lambda^{-1} \\ U_1 \end{pmatrix}\right\|_F = \left\|\begin{pmatrix} \Lambda^{-1} & 0 \\ 0 & I \end{pmatrix}\begin{pmatrix} \Lambda U_2 - I \\ U_1 \end{pmatrix}\right\|_F \leq \frac{1}{\min\left(\lambda_{\min}(\Lambda), 1\right)}\left\|\begin{pmatrix} \Lambda U_2 - I \\ U_1 \end{pmatrix}\right\|_F.$$

Combining (71) by Cauchy-Schwartz inequality we have

$$\|\nabla L_2(\hat{U})\|_F \|\hat{U} - \hat{U}_*\|_F \geq \left\langle \nabla L_2(\hat{U}), \hat{U} - \hat{U}_* \right\rangle \geq L_2(\hat{U}) \geq \frac{\lambda_{\min}(\Lambda)^2}{30d} \|\hat{U} - \hat{U}_*\|_F^2, \tag{73}$$

which yields that

$$\|\nabla L_2(\hat{U})\|_F \geq \frac{\min\left(\lambda_{\min}(\Lambda), 1\right)^2}{30d} \|\hat{U} - \hat{U}_*\|_F. \tag{74}$$

Combine two types of parameterizations to get $\Theta = (\widetilde{U}_{12}^\top, \widetilde{U}_{22}^\top, U_2^\top, U_1^\top)^\top$. Let $\mathrm{Vec}(A)$ be the vectorization operator in row-wise order. For example, $\mathrm{Vec}\begin{pmatrix} 1 & 2 \\ 3 & 4 \end{pmatrix} = (1, 2, 3, 4)^\top$. Define $W = (u_{-1}U_{12}^\top, u_{-1}U_{22}^\top, u_{-1}U_{21}^\top, u_{-1}U_{11}^\top)^\top$

Then we have

$$\mathrm{Vec}(W) = \begin{pmatrix} I_{d^2} & & & \\ & I_{d^2} & & \\ & -I_{d^2} & I_{d^2} & \\ -I_{d^2} & & -\frac{1}{2}(I_{d^2} + T) & I_{d^2} \end{pmatrix} \mathrm{Vec}(\Theta) =: J \mathrm{Vec}(\Theta). \tag{75}$$

Here $T \in \mathbb{R}^{d^2 \times d^2}$ is the transpose operator. That is $T\,\mathrm{Vec}(A) = \mathrm{Vec}(A^\top)$ for any $d \times d$ matrix $A$. Hence by chain rule we have $\nabla L(\mathrm{Vec}(\Theta)) = J^{-1}\nabla L(\mathrm{Vec}(W))$. Therefore we have

$$\|\nabla L(\Theta)\|_F^2 = \|\nabla L(\mathrm{Vec}(\Theta))\|^2 = \|J^{-1}\nabla L(\mathrm{Vec}(W))\|^2 \leq \|\nabla L(\mathrm{Vec}(W))\|^2 \leq \frac{1}{u_{-1}^2}\|\nabla L(\mathrm{Vec}(U))\|^2 = \frac{1}{u_{-1}^2}\|\nabla L((U))\|_F^2. \tag{76}$$

Adding (69) and (74) we have

$$\begin{aligned}
&\|\nabla L(\Theta)\|_F^2 \\
=&\|\nabla L_2(\hat{U})\|_F^2 + \|\nabla L_1(\widetilde{U})\|_F^2 \\
\geq& \min\left(\frac{\lambda_{\min}(\Lambda)^2}{30d}, \frac{1}{30d}, \frac{\lambda_{\min}(\Lambda)^4}{10}, \frac{1}{10}\right) \left(\|U_1\|_F^2 + \left\|U_2 - \Lambda^{-1}\right\|_F^2 + \|\widetilde{U}_{12} + \Lambda^{-1}\|_F^2 + \|\widetilde{U}_{22} - \Lambda^{-1}\|_F^2\right).
\end{aligned} \tag{77}$$

Combining it with (76), we finally obtain

$$\begin{aligned}
&\|\nabla L((U))\|_F^2 \\
\geq& u_{-1}^2 \min\left(\frac{\lambda_{\min}(\Lambda)^2}{30d}, \frac{1}{30d}, \frac{\lambda_{\min}(\Lambda)^4}{10}, \frac{1}{10}\right) \left(\|U_1\|_F^2 + \left\|U_2 - \Lambda^{-1}\right\|_F^2 + \|\widetilde{U}_{12} + \Lambda^{-1}\|_F^2 + \|\widetilde{U}_{22} - \Lambda^{-1}\|_F^2\right) \\
\geq& \beta^2 \min\left(\frac{\lambda_{\min}(\Lambda)^2}{30d}, \frac{1}{30d}, \frac{\lambda_{\min}(\Lambda)^4}{10}, \frac{1}{10}\right) \left(\|U_1\|_F^2 + \left\|U_2 - \Lambda^{-1}\right\|_F^2 + \|\widetilde{U}_{12} + \Lambda^{-1}\|_F^2 + \|\widetilde{U}_{22} - \Lambda^{-1}\|_F^2\right) \\
=& c\left(\left\|U_{11} + U_{12} + \bar{U}_{22} + \bar{U}_{21}\right\|_F^2 + \left\|U_{22} + U_{21} - \frac{\Lambda^{-1}}{u_{-1}}\right\|_F^2 + \|U_{12} + \frac{\Lambda^{-1}}{u_{-1}}\|_F^2 + \|U_{22} - \frac{\Lambda^{-1}}{u_{-1}}\|_F^2\right).
\end{aligned} \tag{78}$$

$\square$

