# OpenReview forum: "Task Descriptors Help Transformers Learn Linear Models In-Context"
_ICML.cc/2024/Workshop/ICL — ICML 2024 Workshop ICL Poster_

### Official Review · Reviewer_6bo4 · 2024-06-03
**Good paper.**

**Rating:** 3
**Fit:** 3
**Confidence:** 3

**Workshop Review:**

This paper focuses on in-context learning (ICL), the linear regression model via a single-layer Linear Self-Attention  (LSA) model with a task descriptor in the token embedding matrix. They theoretically prove the global convergence of the parameters (to one global minimizer) from a certain initialization, and they empirically show the benefits of a task descriptor. In general, I think this is a good paper that extends the results in Zhang et al. 2023 by showing that a task descriptor can help the ICL of a linear model.

I believe the proof is correct from reading the appendix, and I believe this paper should be accepted by this workshop. Below, I have some suggestions for how to make the paper more complete.

1. There are multiple ways to introduce a task descriptor in the token embedding matrix E_\tau. I think maybe you can say more about why you think the form (1) is a good way to formulate this token matrix. I feel this token embedding matrix is kind of far away from the example in figure 1, since in figure 1, you first give the task description (but only once) and then offer the model some examples, while in (1), it seems that you are giving the task description once for each example. Is there any way to only give the task descriptor once in the token embedding matrix (that is, the \mu_\tau appears once in the matrix E_\tau) ?

2. There is one confusion in Theorem 3.1: you said, 'if n \to \infty, then the parameters converge to XXXX.' There are two limit processes in this statement: n \to \infty and t \to \infty, and exchanging two limit processes does not always give you the same result. You should make it clearer which one you mean: do you mean lim_{n \to \infty} lim_{t \to \infty} W_n(t) = W_* or the lim_{t \to \infty} lim_{n \to \infty} W_n(t) = W_*, or are they actually equivalent?

3. One interesting result in Fig 2 is that the minimal loss for ICL with task descriptor is much smaller than the one without it. I think it is worthwhile to theoretically compute it (since in experiments you are using LSA, this makes sense because the calculation is not hard). If this is the case theoretically, it will strengthen your claim that a task description helps. Also, does the convergence rate hold the same with the case without task descriptors? In Zhang et al,2023, they proved that when facing a new task, the convergence rate is O(1/M + 1/N^2) where M and N are respectively the sequence length for the test task and training task. Does this hold with the task descriptor, or is it faster?

4. The data distribution you are considering seems to be a special case of [1], where \beta^* is a zero vector. In [1], they propose that an additional MLP layer can help solve this ICL problem. With the token embedding matrix (1), can you solve the ICL problem in [1] with only an LSA layer? I think you can also discuss more the relationship between your design and the one in [1] for solving ICL linear regression.
[1] In-Context Learning of a Linear Transformer Block: Benefits of the MLP Component and One-Step GD Initialization

**Reason For Not Giving Higher Score:**

See above.

**Reason For Not Giving Lower Score:**

See above.

---

### Official Review · Reviewer_ZpvJ · 2024-06-08
**Review of submission #35**

**Rating:** 3
**Fit:** 3
**Confidence:** 3

**Workshop Review:**

# Summary
This paper theoretically explores how task descriptors can enhance the in-context learning (ICL) capabilities of transformer models, particularly for linear regression tasks. This paper investigates the role of task descriptions in addition to in-context examples and present both theoretical and empirical results demonstrating the benefits. This paper shows that transformers, when provided with task descriptors, achieve significantly lower loss in ICL tasks. They provide a global convergence theorem and empirical evidence supporting their claims.

# Strength
The authors provide a solid theoretical foundation for their claims, including a global convergence theorem that shows how the trained transformer models achieve lower loss with task descriptors. Incorporating task descriptors into transformer models to improve ICL is already empirically examined on previous works, but less work explore this in a theoretical way.

# Weakness
1. The experiments are primarily focused on linear regression tasks. While this provides clear evidence for the proposed approach, it would be beneficial to see how task descriptors perform in more complex and diverse tasks.
2. The setting in this work is to use mean $u_t$ for each task $t$. I am concerned about this setting, because in classical NLP tasks, the task descriptors typically are not the mean of the input vector. I understand that this setting may works well in the setting of LSA transformer, but it might be helpful to consider a setting that is also aligned with classic ICL setting in NLP. The currently used task descriptors of mean $u_t$ is not aligned with the presented task descriptor you use in figure 1.

# Question
1. The assumption 2.1 and formulation (12) may require re-organized.

**Reason For Not Giving Higher Score:**

N/A

**Reason For Not Giving Lower Score:**

N/A

---

### Meta-Review · Area_Chair_idPj · 2024-06-17

**Recommendation:** 2

**Metareview:**

Reviewers agree this is a nice theoretical extension of Zhang et al. (2023) to the presence of a task descriptor.

---

### Decision · Program_Chairs · 2024-06-17

Accept (Poster)